# Noble Metallic Pyramidal Substrate for Surface-Enhanced Raman Scattering Detection of Plasmid DNA Based on Template Stripping Method

**DOI:** 10.3390/mi12080923

**Published:** 2021-08-02

**Authors:** Wenjie Wu, Rui Li, Maodu Chen, Jiankang Li, Weishen Zhan, Zhenguo Jing, Lu Pang

**Affiliations:** 1School of Physics, Dalian University of Technology, Dalian 116000, China; wuwenjie@mail.dlut.edu.cn (W.W.); xjljk@mail.dlut.edu.cn (J.L.); zhanwsh@dlut.edu.cn (W.Z.); jingzg@dlut.edu.cn (Z.J.); 2School of Materials Science and Engineering, Dalian University of Technology, Dalian 116000, China; panglu@mail.dlut.edu.cn

**Keywords:** template stripping method, adhesive polymer, pyramid, surface-enhanced Raman spectroscopy (SERS), plasmid DNA

## Abstract

In this paper, a new method for manufacturing flexible and repeatable sensors made of silicon solar cells is reported. The method involves depositing the noble metal film directly onto the Si template and stripping out the substrate with a pyramid morphology by using an adhesive polymer. In order to evaluate the enhancement ability of the substrate, Rhodamine 6G (R6G) were used as surface-enhanced Raman scattering (SERS) probe molecules, and the results showed a high sensitivity and stability. The limit of detection was down to $10^{-12}$ M for R6G. The finite-difference time domain (FDTD) was used to reflect the distribution of the electromagnetic field, and the electric field was greatly enhanced on the surface of the inverted pyramidal substrate, especially in pits. The mechanism of Raman enhancement of two types of pyramidal SERS substrate, before and after stripping of the noble metal film, is discussed. By detecting low concentrations of plasmid DNA, the identification of seven characteristic peaks was successfully realized using a noble metallic pyramidal substrate.

## 1. Introduction

Surface-enhanced Raman spectroscopy (SERS) is a potent noninvasive spectroscopy technology that can detect and characterize small organic molecules and large biomolecules at very low concentrations and at a single-molecule level. Molecular identification in a low-concentration sample aqueous solution provides the potential for the unlabeled detection and identification of various analytes [1,2,3,4]. Since the discovery of SERS on the surface of noble metals, a new era of Raman spectroscopy has emerged. Metallic substrates with periodic subwavelength structures exhibit unique surface isobaric properties, providing an enhanced electromagnetic field and producing a strong optical response [5,6,7], resulting in SERS, which is of great scientific significance and considerable technical importance for the development of nanophotonic devices, data storage, and biosensors. As such, noble metallic nanoparticles already have widespread applications in creating local surface plasmon resonance (LSPR). By adjusting the geometry and optical properties of the nanoparticles, the Raman signal is greatly enhanced [8,9,10]. Hence, to control nanostructures and generate a larger local EM field, the design and manufacture of ordered and uniform nanostructured SERS substrates has become critical. In the past thirty years, solid support matrices with SERS effects, such as nanocolloids [11], rough metal surfaces [12], and periodic nanostructures [13,14,15], have been widely reported in the literature [16,17]. However, their limited stability and reproducibility often limit their practical applications [18,19]. Various techniques have been reported for the manufacture of uniform nano-SERS substrates with controllable periods, sizes, and geometries, such as focused ion beam lithography [20], electron beam lithography [21,22,23,24], and nanoindentation [25,26,27]. However, these techniques are expensive, time-consuming, and difficult to reproduce for large quantities [28,29].

In this study, high sensitivity SERS substrates were prepared using a template stripping method. We adopted anisotropic etching of crystalline silicon solar cell substrates as nanostructured surface templates with sharp edges, such as commercially available pyramid arrays [30], deposited a certain thickness of noble metal film in the silicon pyramid gap, and then applied a UV-curable polymer (UVcp) on the surface [31]. UVcp is a kind of transparent colloid that is composed of polyurethane acrylate. Under the irradiation of UV light, the UVcp on the surface of the noble metal film crosslinks to form a network structure to achieve stable binding. After curing, UVcp becomes a flexible polymer, and due to the poor adhesion between the noble metal and the silicon surface, the noble metal film deposited on the silicon template can be easily stripped by UVcp to form a densely arranged nanostructured noble metallic pyramidal substrate (PS) [32,33,34,35]. Due to the stability of silicon, the template is highly reproducible, reusable, and mass-producible [36,37]. The good flexibility of the UVcp used in the stripping of the noble metal film improves the load bearing capacity of the bonding surface [38], the damping characteristics are improved, and stress cracking will not occur at the typical molding stress level [39]. It can not only provide a large angle of curvature to adapt to various surfaces, but also enable the collection of Raman signals in situ because of its transparency [40]. The large Young’s modulus allows the metallic features to avoid deformation during the stripping process and ensures structural integrity.

The structure before stripping of the metal film is called a positive pyramidal substrate (PPS), while the structure after stripping is called an inverted pyramidal substrate (IPS). The bottom gap of the pyramid array and the pyramidal side wall lead to a strong electromagnetic field enhancement, resulting in a rich and uniform SERS hot pots [41]. We carried out a numerical simulation and provide experimental evidence to affirm the excellent sensitivity, outstanding uniformity, and stability of the noble metallic PS, and also applied it to biological detection. As deoxyribonucleic acid (DNA) is an important molecule in biology that carries the genetic information for the growth, development, function, and reproduction of organisms [42], the detection of specific DNA sequences in humans, viruses, bacteria, and plants has becoming vital in various fields [43]; while also being helpful for the early diagnosis and prevention of diseases. In this paper, using Ag/IPS as a SERS substrate, we achieved the low-concentration detection of plasmid DNA and the identification of seven characteristic peaks.

## 2. Materials and Methods

### 2.1. Instrument

SERS signals were gathered by a Raman spectrometer (Renishaw InVia, London, England) with an inverse microscope (Zeiss Axiover 25, Jena, Thuringia, Germany) and several filters (Renishaw InVia, London, England, 532 nm, Cut off: 100 ${\mathrm{cm}}^{-1}$, OD = 8), and SERS spectra were acquired at room temperature. An x-y translational stage was used to adjust the samples with a micron-scale resolution (Renishaw RCH24, London, England). A 532nm laser (75mW, NOVAPRO, RGB Lasersystems, Kelheim, Bavaria, Germany) was employed for the incident light, which was focused onto the sample using a 50× objective (Nikon, MUL03501, Tokyo, Japan) lens. The accumulation time for each spectrum reported was 1 s, and the resolution of all spectra was 1 cm^-1^.

Scanning electron microscopy (SEM) images were taken with an SEM system (SUPARR 55, Zeiss, Jena, Thuringia, Germany). A vacuum magnetron sputtering system (JCP-200, BTSC563, Beijing, China) was used to produce metal nanofilms. UVcp (AA3311, LOCTITE, Shanghai, China) was used for stripping the metallic nanostructures.

Rhodamine 6G (R6G) (R4127-100g, SIGMA, St. Louis, MO, USA) was selected as the probe molecule to test the SERS performance of the PS. The metallic PS were immersed in R6G solutions of different concentrations for 1 h and left to air dry at room temperature for more than 30 min; the samples were used to measure the Raman spectrum. In order to illustrate the biological application of the enhancement substrate, plasmid DNA was chosen as the detected substance.

### 2.2. Preparation of Plasmid DNA to Be Measured

Plasmid DNA was transformed into DH5α competent cells, the bacterial cultures were grown in 5mL LB (Luria-Bertani, Bioworld, Bloomington, IL, USA) broth for 16 h at 37 °C, and finally plasmid DNA was extracted using miniprep (plasmid Miniprep Kit, Axygen, Union City, CA, USA), while the concentration of the plasmid was 120 ng/μL.

### 2.3. Fabrication of Noble Metallic PS

To strip the metallic nanofilm structure onto a flexible substrate, UVcp was used, to obtain a high-fidelity and high-sensitivity sensor. A regular surface textured crystalline silicon solar cell substrate was chosen as the master mold, which placed random pyramids on the surface [44], as shown in Figure 1a; the substrate is about the size of a coin. A noble metal film with a thickness of 200 nm was deposited on the surface of the substrate, as shown in Figure 1b,c, then the UVcp was dropped onto the surface of the noble metal film. Before curing, UVcp as a solvent could flow and fill the PS at nano- and microscales, as shown in Figure 1e. The UVcp was cured with UV irradiation at 365 nm for 30 s at room temperature. Then the UVcp was stripped with tweezers and the noble metallic structure was successfully stripped onto the flexible UVcp substrate. Figure 1d,f show before and after the process of stripping, respectively. The reason for the curling of the nanofilm is its flexibility, due to the soft UVcp substrate. Au and Ag were selected as the noble metals for this work, which resulted in four different substrates: (b) Ag/PPS, (c) Au/PPS, (f) Ag/IPS, and Au/IPS. The process of the proposed method is simple and can realize rapid measurement and protect the substrate surface from oxidation.

### 2.4. Scanning Electron Microscopy Imaging of Pyramidal Structures

To study the morphological characteristics of the PPS and IPS, SEM was used to image the morphology of the nanostructure before and after stripping onto the UVcp substrate. Figure 2 shows top-view SEM micrographs of the four kinds of noble metallic pyramidal substrates at different magnifications. Figure 2a–d shows the high uniformity and evenness of the pyramidal substrates at micron scale, which reflects the stability of the vacuum magnetron sputtering machine. Figure 2e–h shows top-view SEM micrographs of the two noble metal films after successful stripping. Au and Ag films underwent the same stripping process under the same environment. From the enlarged image, the surface of the stripped metal film is seen to be rough and uneven, and not as smooth as before. This is because there is a certain adhesion between the metal film and silicon during the stripping process, which means that the metal film cannot be completely stripped. In addition, the morphology of the IPS obtained by the two noble metals is also different, which is due to the different adhesions of the Au and Ag films to the silicon substrate.

## 3. Results and Discussion

### 3.1. Simulation Analysis of Electric Field Distribution Using PS

To clarify the physical mechanism of the SERS behavior of the noble metallic PS, we used FDTD to simulate the local electric field characteristics of the substrate. A nano-resolution cubic grid in a three-dimensional system was established in the model, including the appropriate boundary conditions. The base model was placed in the air, the periodic boundary conditions close to the model were applied in the x and y directions, and the z-direction was surrounded by a perfectly matched layer (PML), in which a PML with a thickness greater than the wavelength was used to avoid false boundary reflection. The simulation of the regional background refractive index was set to 1, and the data of the Au and Ag material adopted the model data provided by the software itself. The excitation light was an ordinary incident plane wave, whose wavelength was 532 nm, and it was oriented to illuminate the pyramid from the top (along the z-axis), where the polarization of the light was along the x-axis. A 2 × 2 pyramid array was selected as the basic framework. The PPS model was a positive pyramid array, as shown in Figure 3a. The thickness of the metal film was greater than its penetration depth, thus the part covered by the metal film was equivalent to a solid metallic pyramid for the simulation. The IPS model consisted of a noble metallic cuboid containing a hollow pyramid, the x-z view of the model is shown in the Figure 3b. The pyramids in the PPS and IPS models have the same geometric parameters. The side length of each pyramid was 1 μm, and the slope angle θ of the pyramid was 54°, which was determined by the anisotropic corrosion of single-crystal silicon [45]. To avoid transmission due to the thinner thickness at pyramid joints, a 200 nm thick homogeneous metal layer was placed under the pyramid array. Figure 4 shows the two-dimensional electric field distribution diagram of the PPS and IPS along the x–z plane.

Due to the polarization of the incident electric field, there will be two opposite charges on the two adjacent side walls, and they approach each other as they approach the bottom gap, forming a strong localized plasma [46,47]; the maximum strong electric field is at the bottom gap of the adjacent pyramids of PPS. The simulation results showed that the strengthening effect of Ag substrate is stronger than that of Au substrate, because the characteristic dipolar resonances for Ag nanoparticles are below 450 nm, whereas for Au they are located around 550 nm. Thus, a laser wavelength of 532 nm is near resonance for Au, but well above resonance for Ag nanoparticles [48]. Therefore, Au has an absorption peak at 532 nm, which leads to a poor SERS effect for Au substrate.

The influence of pits in the surface of IPS on the enhancement effect cannot be ignored, as shown in Figure 2g,h. The bonding force between Au and silicon is far less than that between Ag and silicon; therefore, during the stripping process, Au film can be completely stripped from the Si substrate, while the morphology is relatively flat. The opposite is true for Ag. Some Ag remains on the template surface, forming more uneven surfaces on Ag/IPS. This surface structure improves the enhancement effect and produces more "hot" sites for molecular detection than smooth conventional substrates.

Since these pits are too small to be measured, we applied some hemispheres to the model in Figure 4d. The radius of the hemispheres is different, which represents the randomness and contingency of the pits on a Ag film surface, as shown in Figure 5a. The radius of hemispheres 1–5 were 20 nm, 25 nm, 30 nm, 35 nm, and 40 nm, respectively, and the distances between adjacent hemispheres were equal. The radius of hemispheres 6–8 was 40 nm, the distance between the adjacent hemispheres was equal and the depression depth was increasing. In line with the actual scale range, the larger the radius of the pit, the better the enhancement effect. Moreover, the deeper the pit, the better the enhancement effect; this is because the deeper pits form a sharper bulge with the side wall of the pyramid, resulting in stronger scattering. The probe molecule can easily reach these regions, Figure 5b illustrates that the enhancement effect of Ag/IPS is better than that of Ag/PPS.

Moreover, the simple nano-particle dimer hotspot is only in the polarization direction of the incident light [49], while the IPS can change the polarization direction of the scattered light, as shown in Figure 6; under the same conditions, the electric field also distributes in the direction perpendicular to the polarization direction of the incident light.

### 3.2. Raman Spectra of R6G on Pyramidal Sensors

In order to make the Raman spectra of probe molecules of all concentrations under the same laser power compare with the threshold range of the spectrum software, the laser defocusing mode with a laser spot of 2~3 μm in diameter was used to make the laser focus on another plane outside the surface of the measured object, so as to reduce the excessive strong spectrum of the measured molecules with high concentration.

To study the SERS sensitivity of the improved structure, the spectra of four substrates were measured using R6G as a probe molecule, with the same concentration of $10^{-6}$ M, Figure 7 shows the SERS sensitivity measurement results obtained from four SERS substrates in R6G. The results show that all four substrates had a SERS effect. The enhancement effect of Ag/IPS was far greater than that of Ag/PPS, which was consistent with the simulation results. It is worth noting that the enhancement effect of Au/PPS was better than that of Ag/PPS, which was contrary to the simulation results. This can be explained by the SPR effect. Previous studies showed that the SPR effect of Au occurs at 515.1 nm and that of Ag is at 395.8 nm, so the SPR effect of Au is stronger than that of Ag [50]. Combined with the simulation, it can be inferred that the enhancement principles of PPS and IPS are different; the enhancement mechanism of IPS is dominated by scattering [51], while that of PPS is dominated by SPR [44].

In addition, there is no characteristic peak (520 ${\mathrm{cm}}^{-1}$) of Si at 520 ${\mathrm{cm}}^{-1}$ with PPS, which proves that the thickness of the noble metal film was greater than its penetration depth, so the enhancement performance of the substrate was not affected by the underlying Si template or UVcp.

To determine the lowest measurable concentration of R6G in the substrate, we collected the Raman signal of R6G solutions with different concentrations ($10^{-9}$ M~$10^{-13}$ M) under the best condition of Ag/IPS, as shown in Figure 8. The corresponding spectra of each concentration were the mean values of Raman peaks from four random locations. An error histogram plotted using the average of the four peaks at 611 ${\mathrm{cm}}^{-1}$ is shown in Figure 9. In this experiment, the SERS signal of R6G solution was easily observed, and the Raman signal intensity increased with the increase of R6G concentration; the lowest detection concentration of the R6G probe on this substrate was as low as $10^{-12}$ M. Obviously, Ag/IPS is very sensitive to SERS effects. The error of the characteristic peaks reflects the homogeneity of the SERS performance of the substrate structure.

Finally, the stability of SERS substrate is very important. The SERS spectrum was measured again for the Ag/IPS soaked in R6G solution with a concentration of $10^{-6}$ M and it was stored it in a 4-inch silicon wafer box and sealed at room temperature for three years. Figure 10 shows the SERS spectrum of the Ag/IPS, the red line is the Raman spectrum after preparation, while the black line is the Raman spectrum of the same substrate after three years. Three years later, the intensity of the Raman peak at 611 ${\mathrm{cm}}^{-1}$ was 29% of that of the initial result, which indicates that the proposed pyramidal SERS substrate has a good stability.

### 3.3. SERS Using Pyramidal Sensors by Detection of Plasmid DNA

The Ag/IPS was prepared by depositing an Ag layer with a thickness of 200 nm on the surface of the template and stripping it with UVcp. Plasmid DNA was used as a probe to detect the Raman signal on Ag/IPS. Due to the extremely short validity period of DNA, the immersion time of the substrate was shortened to 20 minutes. In order to exclude the influence of other extracts, the same enhanced substrate was immersed in LB broth under the same conditions, and its SERS signal was used as the control group; the SERS result is shown in Figure 11a. Furthermore, the intensities of the 681.0, 924.5, 1006.9, and 1412.8 ${\mathrm{cm}}^{-1}$ peaks in the four randomly selected spots are presented in Figure 11b, the average intensities were, respectively, 317.8, 1044.4, 850.3, and 531.8, with 13.12%, 6.59%, 3.51%, and 5.38% variation. Obviously, the intensity of Raman spectra was quite stable, with small fluctuations. Table 1 shows the band positions and their assignments to the spectra of plasmid DNA. Therefore, we can expect to detect plasmid DNA by using the pyramidal SERS substrate made using this method. Here are the meanings of the abbreviations: G: guanine; A: adenine; C: cytosine; T: thymine; Types of vibrations: δ = bending vibrations, $\delta_{s}\;$= scissoring (in–plane bending) vibrations, ν = stretching vibration.

## 4. Conclusions

In summary, by combining the pyramidal template of silicon solar cell substrate, the flexible UVcp, and the noble metal film layer, we proposed a SERS substrate based on a template stripping method. R6G plays the role of the probe molecule, and the SERS results show that these substrates possess good sensitivity and specificity. Combined with the simulation of FDTD and the experimental results, we can explain the different enhancement mechanisms of noble metallic PPS and IPS. During experiments with different R6G concentrations, the Ag/IPS exhibited a good SERS behavior, which indicates that the noble metallic IPS is expected to be an effective substrate for unlabeled SERS detection. Ag/IPS was also used to identify seven major molecular bands of low-concentration plasmid DNA. The low cost, good homogeneity, sensitivity, and stability verify that a noble metallic IPS is a promising candidate for label-free SERS detection in biological applications and mass screening in the future.

## Figures and Tables

**Figure 1 micromachines-12-00923-f001:**
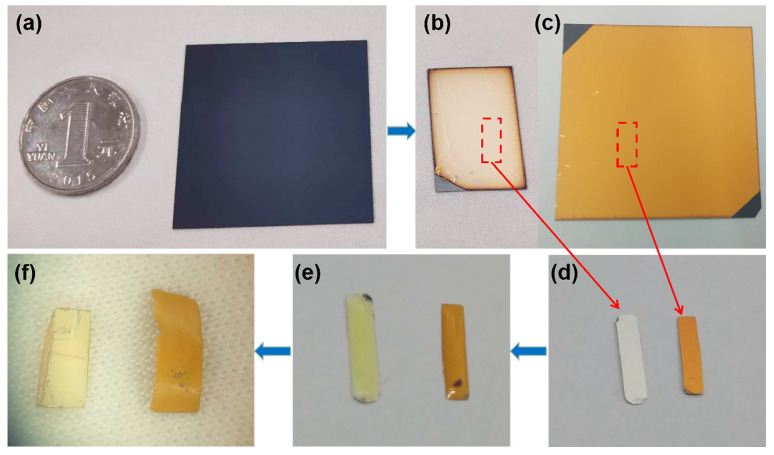
(**a**) Regular surface textured crystalline silicon solar cell substrate; (**b**) Au/PPS; (**c**) Ag/PPS; (**d**) Au/PPS; and Ag/PPS after cutting; (**e**) UVcp was applied to (**d**); (**f**) the Ag/IPS and Au/IPS.

**Figure 2 micromachines-12-00923-f002:**
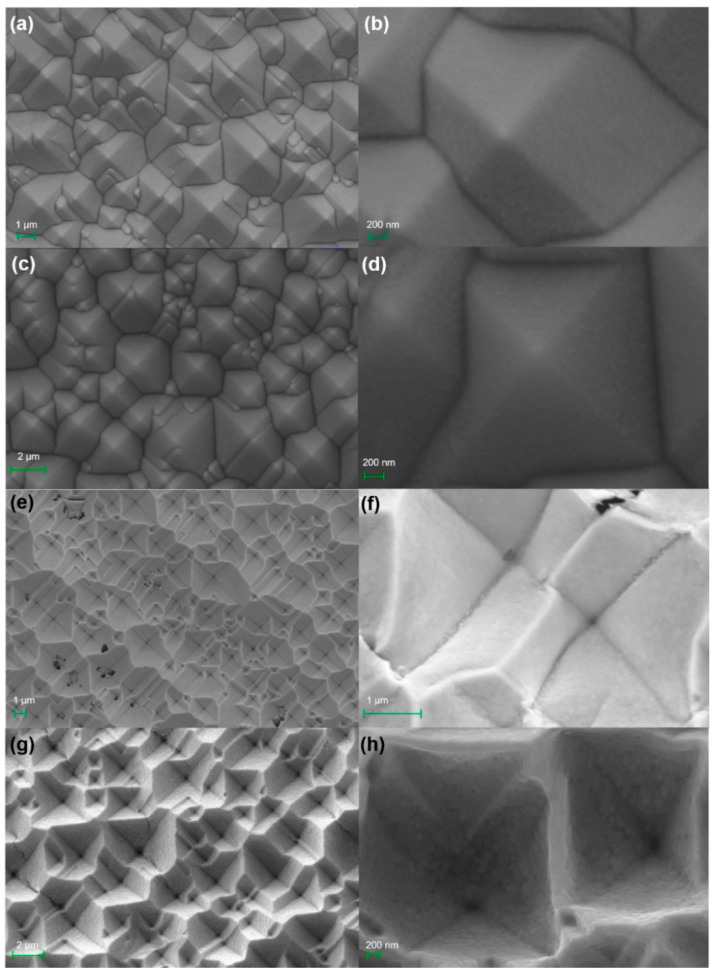
SEM images of four kinds of SERS substrate at different magnifications. (**a**) and (**b**) Au/PPS; (**c**) and (**d**) Ag/PPS; (**e**) and (**f**) Au/IPS; (**g**) and (**h**) Ag/IPS.

**Figure 3 micromachines-12-00923-f003:**
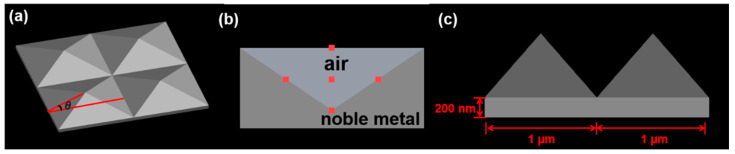
Schematic diagram of the noble metallic nano-pyramidal template. (**a**) PPS model; (**b**) IPS model in the x–z plane; (**c**) side view of the pyramid array.

**Figure 4 micromachines-12-00923-f004:**
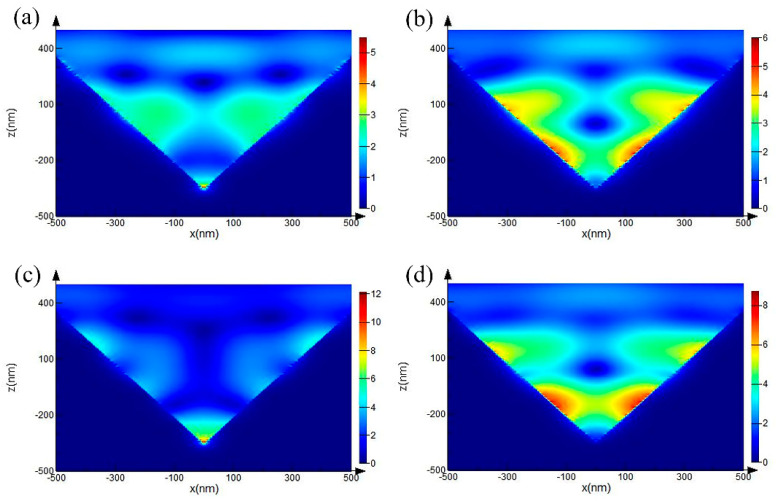
Electric field distribution diagram of a pyramid array in the x-z plane. (**a**) Au/PPS; (**b**) Au/IPS; (**c**) Ag/PPS; (**d**) Ag/IPS.

**Figure 5 micromachines-12-00923-f005:**
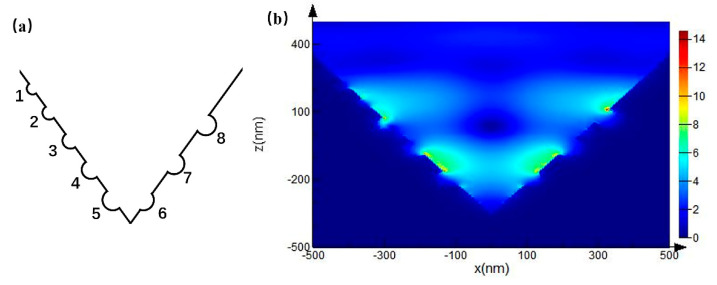
(**a**) Schematic diagram of a pit on the side wall of a pyramid; (**b**) Electric field distribution diagram of Ag/IPS with pits.

**Figure 6 micromachines-12-00923-f006:**
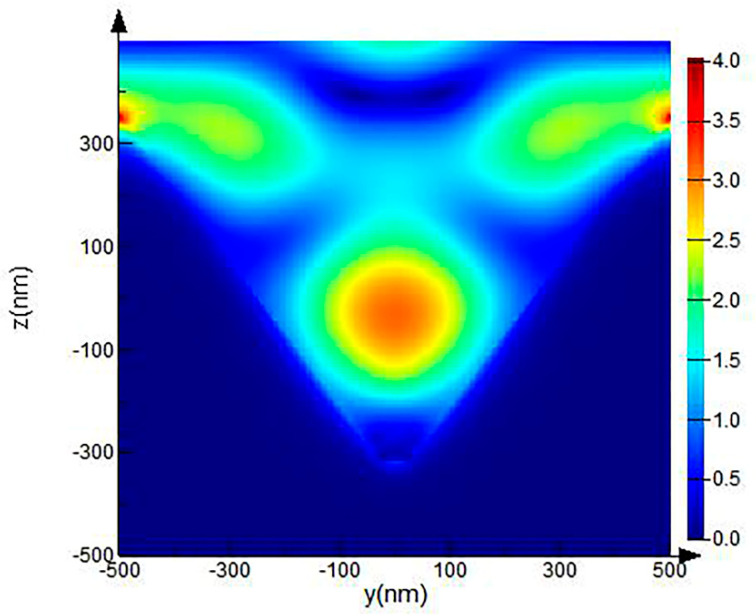
Electric field distribution diagram of Ag/IPS with pits in the y-z plane.

**Figure 7 micromachines-12-00923-f007:**
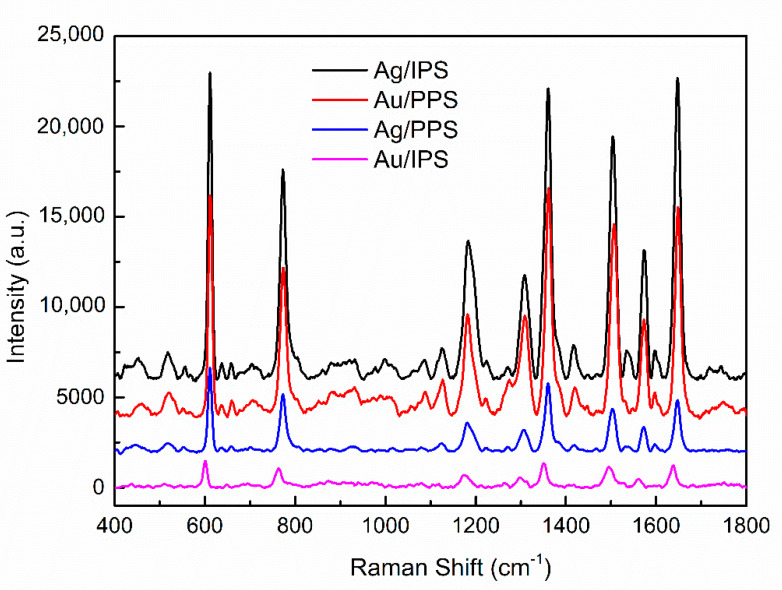
The SERS spectra of R6G ($10^{-6}$ M) gathered on four pyramidal SERS substrates.

**Figure 8 micromachines-12-00923-f008:**
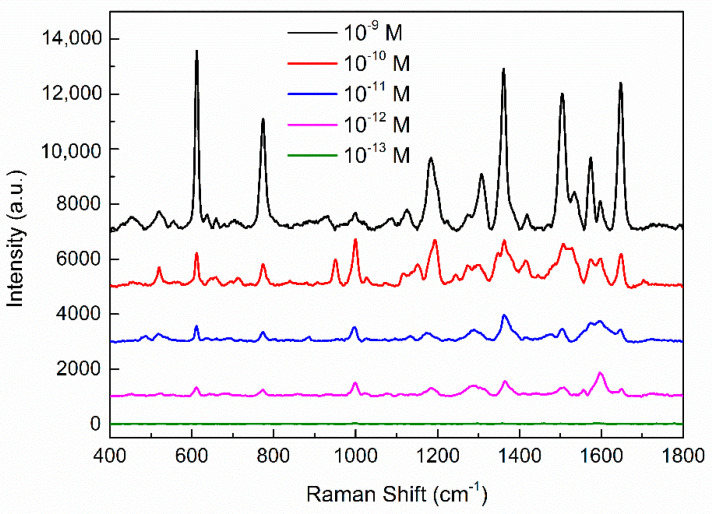
The SERS spectra of various concentrations of R6G solution (ranging from $10^{-9}$ M to $10^{-13}$ M) gathered on Ag/IPS.

**Figure 9 micromachines-12-00923-f009:**
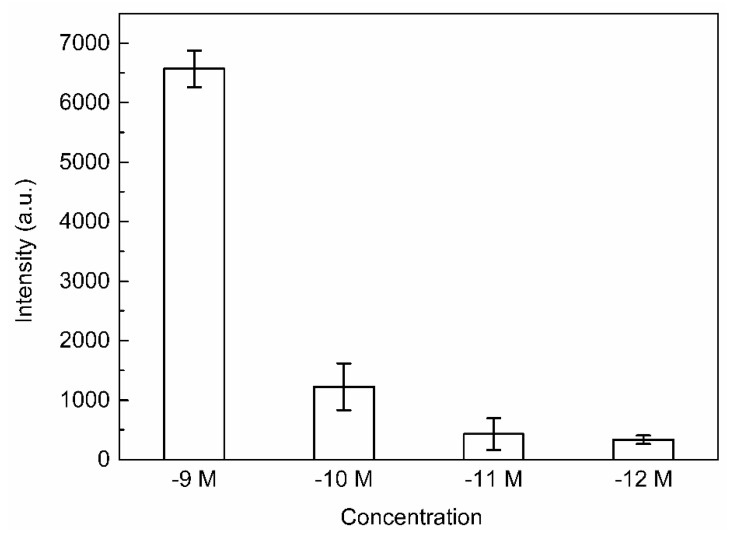
The histogram with errors of the average peak of R6G solution (ranging from $10^{-9}$ M to $10^{-12}$ M) at 611 ${\mathrm{cm}}^{-1}$.

**Figure 10 micromachines-12-00923-f010:**
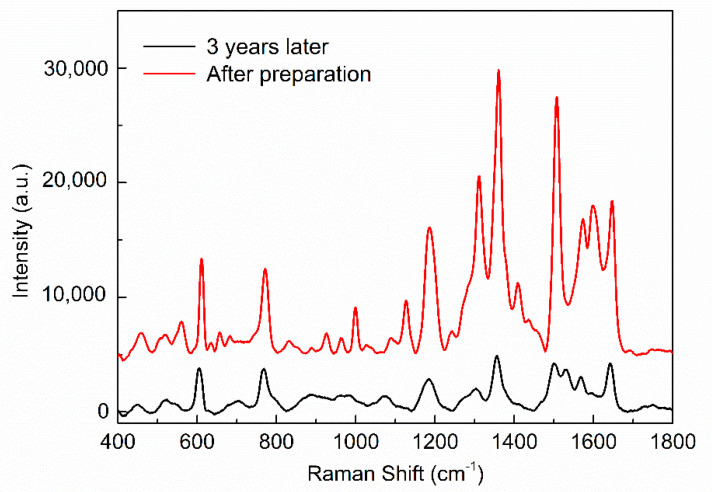
SERS spectra of Ag/IPS after preparation and 3 years later.

**Figure 11 micromachines-12-00923-f011:**
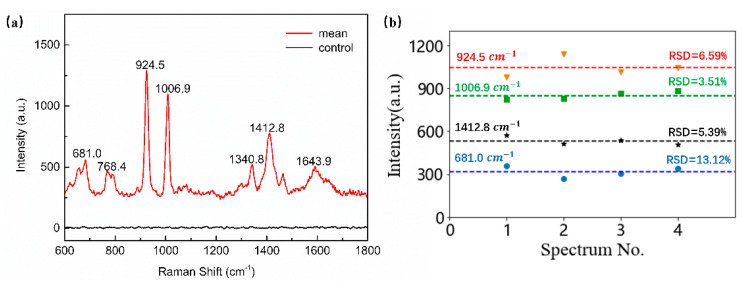
(**a**) The SERS spectra of plasmid DNA and the control group. (**b**) The intensity distributions of the 681.0, 924.5, 1006.9, and 1412.8 ${\mathrm{cm}}^{-1}$ peaks of plasmid DNA in four randomly selected spots on the Ag/IPS, the dotted lines reveal the average intensities of each peak.

**Table 1 micromachines-12-00923-t001:** Band positions and their assignments to spectra of plasmid DNA.

$\mathrm{Raman}\mathrm{Shift}(cm^{-1})$	Spectral Peak Assignment
681.0	682${\mathrm{cm}}^{-1}$-C2’-endo/anti of G [52,53]
768.4	$739\sim 785\;{\mathrm{cm}}^{-1}$-Ring breathing of T [54]
924.5	DNA phosphate skeletal motions [55]
1006.9	1005 ${\mathrm{cm}}^{-1}$-5-methylcytosine [56]
1340.8	1328${\mathrm{cm}}^{-1}$-dT in C2’-endo/anti of S-type [57] 1330${\mathrm{cm}}^{-1}$-ν(C-N) of A, T [58]
1412.8	1412~1417${\mathrm{cm}}^{-1}$-δ(CH),δ(NH), ν(CN) [57]
1463.9	1463 ${\mathrm{cm}}^{-1}$-$\delta_{s}$($N_{3}$-H), δ($C_{5}$-${\mathrm{CH}}_{3}$) of dT [59]

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
