# Peer review of "Noble Metallic Pyramidal Substrate for Surface-Enhanced Raman Scattering Detection of Plasmid DNA Based on Template Stripping Method"

_micromachines, 2021, doi:10.3390/mi12080923_

Round 1
Reviewer 1 Report
In this manuscript, Wu and coworkers reported a new method for flexible and repeatable SERS sensors based on silicon solar cell substrate. The as-prepared Ag/IPS showed a high SERS sensitivity and stability, and can achieve the low-concentration detection of the plasmid DNA. The work is well organized, I recommend it to be published by Micromachines. Below are my comments:
- For the finite-difference time-domain simulations, it seems that the authors used same pyramidal models for both PPS and IPS. However, PPS is convex, while IPS is concave. The authors may need to revise this result.
- Why was the enhancement effect of Ag/IPS far greater than that of Ag/PPS? More discussions are needed.
- Most relevant literature may be cited, such as, Nature, 2010, 464, 392; ACS Nano 2016, 10, 2607
Author Response
Dear reviewers,
Thank you very much for giving us the wise comments to revise our manuscript, the response to the comments is in the attachment.
Wenjie Wu

Reviewer 2 Report
Dear authors,
A method for mass-producing high performance SERS substrates is a very important topic. Your manuscript is a timely contribution to this area. There are a couple of questions and comments.
- Please add more experimental details
- Line 86: what kind of filters were used for what purpose?
- Line 94: after the substrate is immersed into R6G solution, was it simply blow-dried or washed with liquid such as water and allowed to dry?
- Line 106: who manufactures the solar cell you used?
- Line 211: what is the laser spot size after defocusing?
- About references
- History of SERS is too long to be covered by only two references ([11, 12] in Line 41). Please add some more, including earlier works and some review papers.
- The same goes for references for electron beam lithography (Line 45) and production problem (Line 46).
- The detection limit of 10-12 M for R6G is quite remarkable. Recently we have evaluated a number of commercial substrates such as SERitive, Randa S, Wavelet, Silmeco, Q-SERS, and PiCO as well as our own substrates. Some of them, mostly gold, failed to detect 10-6 M R6G with excitation wavelengths of 532 and 633 nm under measurement conditions similar to what you describe. Even silver substrates exhibited detection limits not much lower than 10-7 We used a relatively low excitation intensity of much less than 1 mW to prevent rapid degradation of spectra upon irradiation and build up of a large background. You have explained that the presence of pits (Lines 186~192) as playing an important role on the enhancement effect, but a lot of substrates reported in the literature as well as commercial substrates have tiny defect-like pits, but that does not seem to play such a dominant role, certainly not enough to achieve the detection limit of 10-12 M. I imagine that pyramidal features of your substrate would lead to strong hot spots either at the peak or at the very bottom where crystalline surfaces meet, but your FDTD calculations do not indicate that. I wonder if the superior detection limit you report was also assisted by some unique measurement technique you have devised.
- Your Figure 10 shows that Ag/IPS is still quite active after exposed to air for three years. To put it mildly, it is hard to believe it because none of the commercial Ag substrates and our own Ag substrates retains its activity in the atmosphere after one month at the very best, more likely a week or two though a substrate already exposed to the target molecule can be more stable. As long as we are discussing the long-term stability of substrates before use, data in Figure 10 may not have much relevance in real applications. I would also like you to give more details to your method of protection described in Line119.
Form a technical point of view, the manuscript should be fine after modifications recommended above, and I will leave it up to the editor for the final decision.
Author Response

(The authors gave the same response as above.)
